# Review of Machine Learning in Lung Ultrasound in COVID-19 Pandemic

**DOI:** 10.3390/jimaging8030065

**Published:** 2022-03-05

**Authors:** Jing Wang, Xiaofeng Yang, Boran Zhou, James J. Sohn, Jun Zhou, Jesse T. Jacob, Kristin A. Higgins, Jeffrey D. Bradley, Tian Liu

**Affiliations:** 1Department of Radiation Oncology, Emory University, Atlanta, GA 30322, USA; jing.wang2@emory.edu (J.W.); xyang43@emory.edu (X.Y.); boran.zhou@emory.edu (B.Z.); jun.zhou@emory.edu (J.Z.); kristin.higgins@emory.edu (K.A.H.); jeffrey.d.bradley@emory.edu (J.D.B.); 2Department of Radiation Oncology, Virginia Commonwealth University, Richmond, VA 23219, USA; modeljjy@gmail.com; 3Division of Infectious Diseases, Department of Medicine, Emory University, Atlanta, GA 30322, USA; jtjacob@emory.edu

**Keywords:** lung ultrasound, machine learning, deep learning, COVID-19, artificial intelligence (AI)

## Abstract

Ultrasound imaging of the lung has played an important role in managing patients with COVID-19–associated pneumonia and acute respiratory distress syndrome (ARDS). During the COVID-19 pandemic, lung ultrasound (LUS) or point-of-care ultrasound (POCUS) has been a popular diagnostic tool due to its unique imaging capability and logistical advantages over chest X-ray and CT. Pneumonia/ARDS is associated with the sonographic appearances of pleural line irregularities and B-line artefacts, which are caused by interstitial thickening and inflammation, and increase in number with severity. Artificial intelligence (AI), particularly machine learning, is increasingly used as a critical tool that assists clinicians in LUS image reading and COVID-19 decision making. We conducted a systematic review from academic databases (PubMed and Google Scholar) and preprints on arXiv or TechRxiv of the state-of-the-art machine learning technologies for LUS images in COVID-19 diagnosis. Openly accessible LUS datasets are listed. Various machine learning architectures have been employed to evaluate LUS and showed high performance. This paper will summarize the current development of AI for COVID-19 management and the outlook for emerging trends of combining AI-based LUS with robotics, telehealth, and other techniques.

## 1. Introduction

The COVID-19 pandemic has posed an extraordinary challenge to the global public health system due to the high infection and mortality rate [1]. The hallmark of severe COVID-19 is pneumonia and acute respiratory distress syndrome (ARDS) [2,3,4]. Among symptomatic patients with COVID-19, 14% are hospitalized, 2% require intensive care with an overall mortality rate of 5%. Severe illness can occur in healthy individuals but is more frequent among those with common medical morbidities, including increasing age, diabetes and chronic lung, kidney, liver or heart disease, and mortality may be up to 12-fold in these populations [5]. Medical imaging provides an important tool to diagnose COVID-19 pneumonia and reflect the pathological conditions of the lung [6,7,8,9,10,11,12,13]. Lung ultrasound (LUS) or point-of-care ultrasound (POCUS) is an emerging imaging technique that has demonstrated higher diagnostic sensitivity and accuracy than a chest X-ray and is comparable to CT in COVID-19 diagnosis [14]. For simplicity, we will use LUS instead of LUS/POCUS in this paper. LUS has unique advantages of being portable, prompt, repeatable, low cost, easy to use, and free of ionizing radiation [15,16]. LUS can be used at all steps to evaluate COVID-19 patients from triage, diagnosis, and follow-up exams. Over two years into the COVID-19 pandemic, the number of daily confirmed COVID-19 cases is still striking, and new variants are being identified, requiring efficient diagnostic tools to guide clinical practice in triage and management of potentially suspected populations [17,18,19]. Figure 1 shows the rapid increases of artificial intelligence (AI) publications for COVID-19 diagnosis with US and all imaging modalities (CT, X-ray, and US) in PubMed database.

This review aims to outline current and emerging clinical applications of machine learning (ML) in LUS during the COVID-19 pandemic. We conducted a literature search with keywords including “COVID-19”, “AI”, “Machine Learning”, and “Deep Learning”. We first searched the PubMed database with a combination of “COVID-19”AND “ultrasound” AND “AI OR Machine Learning OR Deep Learning OR Auto” in the title and abstract fields. For a more comprehensive literature review, we further searched keywords: “COVID-19” + “Ultrasound” + “AI/Machine Learning/Deep Learning” on Google Scholar. Combining the results from PubMed and Google scholar, we reviewed recent original research articles focusing on ML applications of LUS in COVID-19, consisting of journal articles, conference papers, arXiv or TechRxiv preprints and book chapters. More than 35 research articles were reviewed and summarized in the following sections.

## 2. LUS Scan Protocols and Features of COVID-19

### 2.1. LUS Scan Protocols

A variety of LUS scanning schemes for COVID-19 patients exist. LUS has been performed with US scanners ranging from high-end systems to hand-held devices [20,21,22,23,24,25] and different transducers, including convex, linear, and phased array transducers [26,27,28,29]. Scans have been performed with patients in the sitting, supine or decubitus positions. Several scan methods have been reported, including the BLUE protocol (6-point scans), 10-point scans [30,31], 12-zone scans [32,33], and 14-zone scans [34,35], all aiming to examine the whole lung in a standardized and thorough manner. The intercostal scanning tries to cover more surfaces in a single scan and evaluates the ultrasound patterns bilaterally in multiple regions to evaluate the overall severity of lung diseases. The BLUE (Bedside Lung Ultrasound in Emergency, Figure 2) protocol works in respiratory failure settings [36] and is thus a suitable practice for rapid COVID-19 scans [37]. Three standardized points for scanning at one side of the lung and bilaterally six key zones in total are examined [37,38]. The three BLUE-points include at one side the upper and lower BLUE-point (anterior) and the PLAPS (posterolateral alveolar and/or pleural syndrome) BLUE-point. Such scanning protocols may be followed to acquire at-home LUS [39].

### 2.2. LUS Image Features for COVID-19 Pneumonia

When compared to liquids or soft tissues, the air-filled lung presents significant differences in acoustic properties. As shown in Figure 3, LUS images distinguish a healthy lung from a lung with interstitial pathology mainly via artifacts [41]. For a healthy lung with a normally aerated pleural plane, the incident ultrasound pulses are almost completely reflected by the regular pleural plane, resulting in horizontal linear artifacts parallel to the pleural plane (A-lines). When the ratio of air to fluid in the lung is changed, the lung and pleural tissues lose the regular structure and can no longer function as a complete specular reflector of the incident ultrasound signals, thus producing various artifacts [29]. One important artifact is the B-line, a type of vertical artifact beginning from the plural plane, focal or confluent [20]. B-lines are believed to correlate with the volume of extravascular lung water and interstitial lung diseases, cardiogenic and non-cardiogenic lung edema, interstitial pneumonia, and lung contusion [42]. Overall, COVID-19 pneumonia/ARDS is associated with the sonographic appearances of pleural line irregularities and B-line artefacts, which are caused by interstitial thickening and inflammation, and increase in number with severity.

Many review papers have discussed the LUS image characteristics for COVID-19 pneumonia [6,14,20,43,44,45,46,47]. When diagnosing COVID-19, the diffuse B-lines are the most commonly seen in US findings, followed by the pleural line irregularities and subpleural consolidations as the next most frequent findings [20,30,31,39,48,49], while the pleural diffusions appear less often [20,21,26,30,31,48,50]. The reappearance of A-lines is expected during the recovery phase or in normal lungs [24,48], and the “white lung” (multi B-lines coalescing) is observed as pneumonia deteriorates [31,45].

### 2.3. LUS Grading Systems of COVID-19 Pneumonia

Various grading systems have been proposed to classify COVID-19 pneumonia using LUS images (A-lines, B-lines, pleural lines, and subpleural consolidates). For instance, Soldati et al. [35] proposed a scoring scheme that classifies the severity of pneumonia into four stages: score 0 is when a regular pleural line and horizontal A-lines are observed, indicating a normal lung; score 1 is when indented pleural lines and vertical white areas below the indent are observed; score 2 is when the broken pleural lines, small-to-large consolidations and multiple or confluent B-lines are observed (white lung happens at this stage); score 3 is the highest level of severity, where the white lung is dense and extends to large areas. Similar scoring systems were proposed by Volpicelli et al. [52], Deng et al. [53], Lichter et al. [54], and Castelao et al. [50]. Such LUS scores could be utilized in AI algorithms for a future auto-grading system of pathology severity.

## 3. Machine Learning in COVID-19 LUS

### 3.1. Public-Accessible Databases

AI is commonly used in medical imaging for detecting diseases [55]. Machine learning (ML) such as deep learning (DL) are powerful techniques used in AI, and large databases are critically important. Many of the current COVID-19 imaging datasets consist of CTs, such as COVID-CT [56], or based on X-rays, such as COVIDx [57], with only a few containing collections of ultrasound images [58]. Table 1 summarizes publicly accessible LUS databases.

Born et al. released a lung ultrasound dataset (POCUS) in May 2020. The POCUS dataset had 1103 images sampled from 64 videos, consisting of 654 COVID-19, 277 bacterial pneumonia, and 172 healthy controls [59], which later was enlarged to include 106 videos [60]. In January 2021, Born et al. released an updated version of the dataset, including 202 videos of COVID-19, bacterial pneumonia, non-COVID-19 viral pneumonia patients and healthy controls [61]. In March 2020, Soldati et al. proposed an internationally standardized acquisition protocol and 4-level scoring schemes for LUS in COVID-19 [35]. They shared 30 positive COVID-19 cases in an online database (ICLUS-DB database), containing about 60,000 frames. Roy et al. proposed an extended and fully-annotated version of the previous ICLUS-DB database, which includes a total of 277 LUS videos from 35 patients [62]. The frames from these videos were labeled according to the 4-level scoring system.

More recently, in March 2021, Ebadi et al. presented an open-access LUS benchmark dataset, COVIDx-US, collecting images from multiple sources [63]. The COVIDx-US dataset consists of 150 videos (12943 frames) in total, categorized into four subsets: COVID-19, non-COVID-19, other lung diseases, and healthy patients. Another group worked on a 12-lung-field scanning protocol of thoracic POCUS (T-POCUS) images for COVID-19 patients [64]. The preliminary dataset consists of 16 subjects (mean age 67 years old), with 81% being male. Their data were stored in the Butterfly IQ cloud, so it might not be openly accessible.

### 3.2. Traditional Machine Learning Classifiers

Several traditional ML algorithms were utilized for LUS image analyses. Principal component analysis (PCA) and independent component analysis (ICA) could extract image features. Random forest, support vector machine (SVM), k-Nearest Neighbour (KNN), decision trees, and logistic regression could perform subsequent classifications [65]. In COVID-19 LUS image analyses, SVM classifiers are commonly used.

#### 3.2.1. Support Vector Machine (SVM) Classifier

Carrer et al. trained a support vector machine (SVM) classifier to predict the COVID-19 LUS score on subsets of the ICLUS-DB database [66]. The process mainly focused on the pleural line and areas beneath the line, which was implemented in two major steps. The first step was to identify the pleural line in each frame picked from the LUS videos. A circular averaging filter was used to smooth out the background noise. Then a Rician-based statistical model [67] was employed to enhance the bright areas (plural lines and other tissues) from the dark background. Next, a local scale hidden Markov model (HMM) and the Viterbi algorithm (VA) [68] were adapted to detect all the horizontal linear high-lighted sections. The deepest linear sections with strong intensities connected to the identified “pleural line.” Once the pleural line was reconstructed, the second step was to train the SVM classifier on eight parameters extracted from the geometric and intensity properties of the pleural line and the areas below the line. Such supervised classification required a smaller dataset compared to a deep neural network and was computed much faster. The labeling used a 4-level scoring system [35], from 0-healthy to 3-severe. The overall accuracy in identifying the pleural lines was 84% for convex and 94% for linear probes. The accuracy of the auto-classification using SVM was about 88% for convex and 94% for linear probes.

Wang et al. extracted features of the pleural line (thickness, roughness, mean and standard deviation of pleura intensities) and B-lines (number, accumulated width, attenuation, and accumulated intensity) from 13 moderate, seven severe, and seven critical cases of COVID-19 pneumonia. For a binary comparison of severe/non-severe, several features showed statistically significant differences. An SVM classifier yielded a high area under the curve (AUC) of 0.96, with a sensitivity of 0.93 and specificity of 1 [69]. One could also separate the evaluation into two steps of feature extraction and image classification [70]. Four CNN models (Resnet18, Resnet50, GoogleNet, and NASNet-Mobile) were tried to extract features from LUS images and then fed the features to the SVM classifier. For a dataset of 2995 images (988 COVID-19, 731 Pneumonia, and 1276 Regular), SVM classifiers reached exceptional results of accuracy, precision, recall, and F1-score (most exceeding 99%), with all four set of features extracted by the four CNN models.

#### 3.2.2. Artificial Neural Network (ANN) Classifier

Raghavi et al. tried the Hopfield neural network (a type of artificial neural network for human memory simulation) to classify a dataset with 266 positive COVID-19 and 499 negative cases (training/test: 80/20%), achieving an accuracy of over 83.66% [71].

### 3.3. Deep Learning (DL) Models

Due to the increasing chip processing capability and reduced computational cost, deep learning (DL) has been a fast-developing subset of machine learning. Many DL architectures proposed for COVID-19 detection were based on convolutional neural networks (CNNs), trained with collected LUS datasets.

#### 3.3.1. Convolutional Neural Networks (CNNs)

Born et al. trained a convolutional neural network (POCOVID-Net) on their *POCUS* dataset with 5-fold cross-validation [59]. They combined the convolutional part of VGG-16 [72] with a hidden layer containing 64 neurons with ReLU activation. The trained classifier achieved an accuracy of 0.89, a sensitivity of 0.96, and a specificity of 0.79 in diagnosing COVID-19 pneumonia. Later, they used VGG-CAM (Class Activation Map (CAM)) or VGG to train three-class (COVID-19 pneumonia, bacterial pneumonia, and healthy controls) models on POCUS [60]. Both VGG-CAM and VGG yielded accuracies of 0.90. In January 2021, they extended the dataset and trained a frame-based classifier, yielding a sensitivity of 0.806 and a specificity of 0.962 [61]. Similarly, Diaz-Escobar et al. performed both three-class (COVID-19, bacterial pneumonia, and healthy) and binary (COVID-19 vs. bacterial and COVID-19 vs. healthy) classifications on POCUS, with pre-trained DL models (VGG19, InceptionV3, Xception, and ResNet50) [73]. Their results showed that InceptionV3 worked best with an AUC of 0.97 to identify COVID-19 cases from pneumonia and healthy controls. Roberts et al. tested VGG16 and ResNet18, and their results from VGG16 outperformed the ResNet18 counterparts [74].

For the fully-annotated ICLUS-DB, Roy et al. proposed an auto-scoring process for image frames and videos separately, along with a pathological artifact segmentation method [62]. As for image frames, the anomalies or artifacts were automatically detected with the spatial transformer networks (STN). For two different STN crops taken from the same image, they regularized their scores to be consistent, so the method was named the regularized spatial transformer networks (Reg-STN). As for videos, scores reflect the overall distribution of frames in sequence by employing a softened aggregation function based on uninorms [75]. Their Reg-STN method outperformed all other tested baseline methods for frame-based classifications with an F1-score of 65.1%. Their softened aggregation method outperformed the maximum and average baseline aggregation methods for video-based scoring, with an F1-score of 61 ± 12%. For auto-segmentation evaluation, their model achieved an accuracy of 96% at a pixel level. Yaron et al. tried to improve Roy’s method by fixing the discrepancy between convex and linear probes, noticing the artifacts (“B-lines”) were tilting in convex probe images [76]. By rectifying the convex probe images from the polar coordinate, the artifacts would be axis-aligned like the linear probe, which simplified the DL architecture and yielded a higher F1 score compared to [62] over the same dataset. Reg-STN architectures were also applied on other COVID-19 datasets and achieved satisfactory accuracies of above 0.9 [77].

In another multicenter study, Mento et al. described standardized imaging protocols and scoring schemes in acquiring LUS videos. They trained DL algorithms on 82 confirmed patients and graded videos by aggregation of frame-based scores [78]. The prognostic agreement between DL models [62] and clinicians was 86% in predicting a high or low risk of clinical worsening [78]. Later for a larger dataset of 100 patients, the agreement between AI and clinicians was 82% [79]. Demi et al. applied similar algorithms [78] in a longitudinal study with 220 patients (100 COVID-19 patients and 120 post-COVID-19 patients), and their prognostic agreement between AI and MDs was 80% for COVID-19 patients and 72.5% for post-COVID-19 patients [80].

Ebadi et al. proposed a deep video scoring method based on the Kinetics-I3D network, without the need for tedious frame-by-frame processing [81]. Pneumonia/ARDS features including A-lines, B-lines, consolidation and/or pleural effusion classes were detected from the video and compared with radiologists’ results. They collected a dataset of 300 patients with 100 for each ARDS feature. Five-fold cross-validation results showed high ROC-AUCs of 0.91–0.96 for detecting the three ARDS features (A-lines, B-lines, consolidation and/or pleural effusion).

Since the COVID-19 LUS dataset include both videos and frames, the impact of various training/test splitting scheme should be evaluated [82]. Roshankhah et al. used manually segmented B-mode images and corresponding 4-scale staging scores as the ground truth for 1863 images from 203 videos (of 14 confirmed cases, four suspected cases, and 14 controls). They achieved a higher accuracy of 95% for image-level data splitting (training/test: 90/10%) but a much lower accuracy (<75%) for patient-level data splitting. The overestimation of image-level splitting may originate from the fact that the images from the same patient could be similar but randomly appear in both training and testing subsets.

Another underlying issue is the labeling effort for LUS videos or frames. Durrani et al. investigated the impact of labeling effort by comparing binary classification results from the frame-based method (higher labeling effort) versus the video-based method (lower labeling effort) [83]. They further introduced a third sampled quaternary method to annotate all frames based on only 10% positively labeled samples from the whole dataset, which outperformed the previous two labeling strategies. Gare et al. tried to convert a pre-trained segmentation model into a diagnostic classifier and compared the results from dense vs. sparse segmentation labeling [84]. Tested on a restricted dataset of 152 images from four patients (three COVID-19 positives and one control), they found that with pretrained segmentation weights and dense labeling pretrained U-net, the classifier performs best with an overall accuracy of 0.84.

Considering the development of portable LUS and the need for rapid bedside detection, Awasthi et al. proposed a lightweight DL architecture of COVID-19 LUS diagnosis. The new method, namely Mini-CovidNet, modifies MobileNet with focal loss. Mini-CovidNet obtained an accuracy of 0.83 on the POCUS dataset [59], which is similar to POCOVID-Net. Still, the number of parameters was 4.39 times lesser in Mini-CovidNet, and thus consumed smaller memory, making it appealing to mobile platforms [85]. On the other hand, an interpretable subspace approximation with adjusted bias (Saab) multilayer network was proposed to read LUS images with low-complexity and low-power consumption, which appeals to personal devices [86].

#### 3.3.2. Hybrid Models: Combining CNNs with Other Methods

Hybrid DL algorithms combining backbone CNNs with other units such as the long short-term memory (LSTM) were introduced to improve the model performance. Barros et al. tailored a hybrid CNN-LSTM model to classify LUS videos by extracting spatial features with CNNs and then learning the temporal dependence via LSTM [87]. Their hybrid model reached a higher accuracy of 93% and sensitivity of 97% for COVID-19 cases, compared to other primitive spatial-based models. Dastider et al. used the Italian LUS database (ICLUS-DB) to train a frame-based four-score CNN-based architecture with a backbone of DenseNet-201 [88]. After integrating the LSTM units (CNN-LSTM), the model performance improved by 7–12% compared to the original CNN architecture.

Besides LSTM units, other add-ons such as feature fusion or denoising could also help. A CNN-based classifier with a multilayer fusion functionality per block was also tested to enhance the performance and achieved high metrics of above 0.92 over the *POCUS* dataset [89]. Che et al. utilized a multiscale residual CNN with feature fusion strategy to evaluate a dataset consisting of both POCUS dataset and ICLUS-DB and obtained an average accuracy of 0.95 [90]. A spectral mask enhancement (SpecMEn) scheme was introduced to reduce the noise in the original LUS images; thus, the SpecMEn improved the accuracy and F1-score of DL models (CNN variants) by 11% and 11.75% on the POCUS dataset [91].

For COVID-19 diagnosis, plural lines and sub-plural symptom features in LUS are signatures of pneumonia severity [92], so some DL algorithms took advantage of the identification and segmentation of pleura, A-lines, and B-lines to improve pneumonia assessment. Baloescu et al. developed deep CNN to detect B-lines from 400 ultrasound clips and evaluate COVID-19 severity [93]. For binary classification of presence or absence of B-lines, they reached a sensitivity of 0.93 and a specificity of 0.96. For multiscale severity based on B-lines, the model reached a weighted kappa of 0.65. Panicker et al. presented a method to first detect the pleura via extracted acoustic wave propagation features; after obtaining the region below pleura, the infection severity was classified with VGG-16 from input regions [94]. They achieved an accuracy, sensitivity, and specificity of 0.97, 90.2, and 0.98, respectively, for 5000 video frames from ten patients over their infection to the full recovery phase. Considering the complexity of pathology behaviors for COVID-19, Liu et al. built a new LUS dataset with multiple COVID-19 symptoms (A-line, B-line, P-lesion, and P-effusion), namely COVID19-LUSMS, consisting of 71 patients [95]. They presented a semi-supervised two-stream active learning (TSAL) method with multi-label learning, and achieved high accuracies greater than 0.9 for A-line, B-line, and moderate accuracies greater than 0.8 for P-lesion and P-effusion. Such aid of artifacts detection improved the classification performance of the neural network.

One encouraging point about these LUS artifacts, such as B-lines, is that though they are related to pathology in various lung diseases, deep learning techniques could identify COVID-19 from other types of pneumonia [96]. Over a combined B-lines dataset consisting of 612 LUS videos from 243 patients with COVID-19, non-COVID acute respiratory distress syndrome, and hydrostatic pulmonary edema, the trained CNNs (Xception architecture) classifiers showed AUCs of 1.0, 0.934, and 1.0, respectively, much better than physician differentiation ability between these lung diseases [97].

#### 3.3.3. Multi-Modality Data and Transfer Learning

Given several datasets of COVID-19 medical images are now available, the amount of total LUS data is still highly limited due to the short time of collecting images since the outbreak. To address the problem of scarcity of available medical images, multi-modality data was utilized to elevate model accuracy for small datasets.

In effort to treat the heterogeneous and multi-modality medical information, a dual-level supervised multiple instance learning module (DSA-MIL) was proposed to fuse the zone-level signatures to patient-level representations [98]. A modality alignment contrastive learning unit combined the representations of LUS and clinical information (such as age, disease history, respiratory, fever, and cough). A staged representation transfer (SRT) scheme was used for subsequent data training. Combining both modalities (LUS and clinical information), they achieved an accuracy of 0.75 for four-level severity scoring and 0.88 for the binary severe/non-severe classification. On the other hand, Zheng et al. built multimodal knowledge graphs from fused CT, X-ray, ultrasound, and text modalities, reaching a classification accuracy of 0.98 [99]. A multimodal channel and receptive field attention network combined with ResNeXt was proposed to process multicenter and multimodal data and achieved 0.94 accuracy [100].

Horry et al. searched COVID-19 chest X-rays [101], CT scans (COVID-19 and non-COVID-19) [56], and ultrasound images (COVID-19, bacterial pneumonia and normal conditions) [59]. They compiled them into a multimodal imaging dataset for training [102]. To minimize the effect of sampling bias introduced by any systematic difference in pixel intensity between datasets, histogram equalization to images using the N-CLAHE method described by [103] was applied. They tried to achieve a reliable classification accuracy with transfer learning to compensate for the limited size of the sample datasets and accelerate the training process. Eight CNNs based models were tested: VGG16/VGG19, Resnet50, Inception V3, Xception, InceptionResNet, DenseNet, and NASNetLarge. As a result, the ultrasound mode yielded a sensitivity of 0.97 and a positive predictive value of 0.99 for normal cases; and a sensitivity of 1.0 and a positive predictive value of 1.0 for distinguishing COVID-19 from the bacterial pneumonia patients.

Karnes et al. explored a novel vocabulary-based few-shot learning (FSL) visual classification algorithm that utilized a pre-trained deep neural network (DNN) to compress images into features and further processed them to vocabulary and feature vector [104]. The knowledge of a pretrained DNN was transferred to new applications, and then only a few training parameters were needed, so the training dataset was largely reduced. Salvia et al. employed residual CNNs (ResNet18 and ResNet50), transfer learning, and data augmentation techniques for multi-level pathology gradings (four main levels and seven sub-levels) over a dataset of 450 patients [105]. For both ResNet18 and ResNet50, and for both four-level and seven-level classifications, the metrics were all above 0.97. By taking advantage of transfer learning, their model could classify the COVID-19 cases even with limited LUS data input. 

A summary of the discussed AI research articles is listed in Table 2.

## 4. Challenges and Perspectives

AI-based medical imaging has shown great potential in evaluating COVID-19 pneumonia [106,107], which could possibly be integrated within a user interface (UI) for decision support and report generation [108]. LUS is a safe, cost-effective, and convenient medical imaging modality, which has demonstrated the potential to serve as the first-line diagnosis tool, particularly in resource-challenged settings. The blooming of ML and DL on LUS is encouraging. Several ML architectures have been developed to differentiate and grade the COVID-19 in a standardized and accurate manner. Therefore, ML algorithms may gain growing value and set a new trend on shaping the LUS into a more reliable and automated tool for COVID-19 evaluation. The high accuracy auto-detection will largely save the clinicians time and effort in decision making and help to reduce possible interobserver errors.

Though many AI publications focused on binary classifications to differentiate COVID-19 from normal or other pneumonia cases, a few multiscale ML classifiers aimed to score stages of pneumonia and diagnose pathology severity quantitatively [62,78,79,80,81,82,88,93,98,105]. These severity scores could be used in the triage and management of patients in clinical settings. Moreover, LUS can be routinely scanned if needed, without the risk of radiation exposure [16] and the burden of an over-complicated disinfection process [109]. Thus, these scores could also be used for monitoring the progress of the disease. It would be of interest to see more attempts on AI severity scoring from LUS and subsequent AI-enabled monitoring and triage workflow of COVID-19 patients.

Though AI-based LUS has many great advantages in the current COVID-19 pandemic, some challenges should be addressed. First, ultrasound poses a major limitation of the incapability of sound waves to penetrate deep into the lung tissues [23], so the deep lesions inside the lung cannot be properly reflected and reconstructed via LUS [110]. Besides poor penetration of sound wave signals for lung tissues, LUS results can be susceptible to the expertise of operators and thus induce low inter-annotator agreement [62,111]. Though ML and DL models aim to diagnose LUS free from human intervention, the training sets still require manual labeling, which may undermine the results. Cautions need to be taken during the labeling phase.

Another limitation comes from the dataset. It is difficult to find annotated LUS images from a large population of patients. This is because not only the online LUS datasets are scarce, but also the annotating process is time-consuming. Though many DL methods were proposed to address the problem, such as transfer learning on multimodal data, the generality of trained models is questionable with intrinsically small datasets. Besides the size of training datasets, the quality of input images could also impact the performance of ML algorithms. A recent study found that the performance of a previously validated DL algorithm worsened on new US machines, especially for a lower IQ, hand-held device [112]. Thus, all researchers are encouraged to contribute to large and comprehensive datasets. Despite these limitations, AI-based LUS is still highly valued in fast and sensible scans.

Nowadays, lightweight, portable ultrasound and telehealth are widely explored during the COVID-19 pandemic. Some studies use AI-robotics to perform tele-examination of patients [113,114], including a telerobotic system to scan LUS on a COVID patient [115]. These robot-assisted systems can increase the distance between sonographers and patients, thus minimizing the transmission risk [116]. In addition, Internet of Things (IoT) technologies are integrated with cloud and AI to monitor and prognose COVID-19 disease [117,118]. In such cases, integration of AI-LUS with telerobotic systems and IoT could automate the process from acquiring imaging data to pneumonia diagnosis and disease monitoring, which will largely increase the efficiency of healthcare systems. Such AI-assisted clinical workflow promises great potential in the context of COVID-19 or other pandemics in the future.

## 5. Conclusions

AI-based LUS is an emerging technique to evaluate COVID-19 pneumonia. To improve the diagnosis efficiency of LUS, scoring systems and databases of LUS images are built for training ML models. An increasing number of ML architectures have been developed. They were able to achieve fairly high accuracy to differentiate the COVID-19 patients from both bacterial-related and other pneumonia cases and grade the pathology severity. In the future, with the increase of LUS datasets, more reliable AI algorithms could be developed and potentially help to diagnose and monitor viral pneumonia to reduce the tremendous burden to the global public health system.

## Figures and Tables

**Figure 1 jimaging-08-00065-f001:**
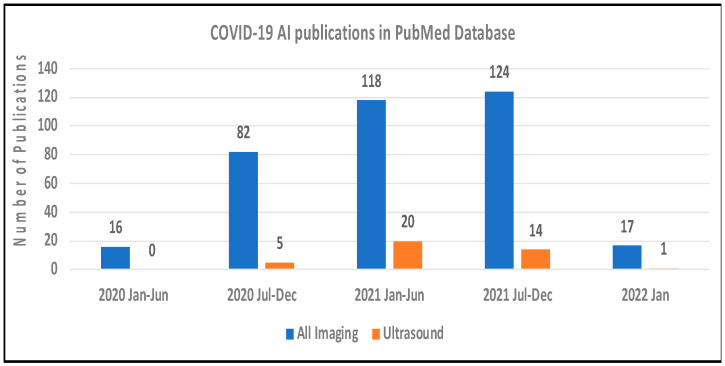
The number of COVID-19 AI publications based on ultrasound (orange bars) and CT, X-ray and ultrasound combined (blue bars) in PubMed Database, as of 17 January 2022.

**Figure 2 jimaging-08-00065-f002:**
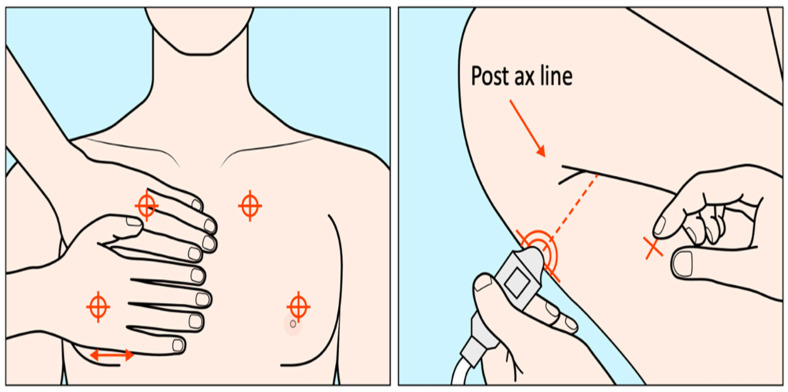
The three standard BLUE points are illustrated (two anterior and one posterior) [modified from [40]]. Two hands are placed on the front chest such that the upper hand touches the clavicle, and the upper anterior BLUE-point is in the middle of the upper hand, while the lower anterior BLUE-point is in the middle of the lower palm. The PLAPS-point is vertically at the posterior axillary line and horizontally at the same level of the lower anterior BLUE-point.

**Figure 3 jimaging-08-00065-f003:**
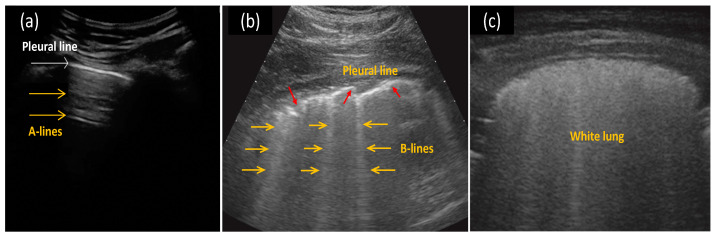
Examples of (**a**) horizontal A-lines (yellow arrows) in a normal lung, (**b**) multiple B-lines (yellow arrows) with an irregular pleura line (red arrows) in a COVID-19 indicative lung, and (**c**) white lung (completely diffused B-lines) for severe COVID-19 pneumonia. Reprinted with permission from ref. [34,51]. Copyright 2020 John Wiley and Sons.

**Table 1 jimaging-08-00065-t001:** Large online open databases of COVID-19 ultrasound images.

Database	Data Characteristics	Access Link
POCUS dataset [59]	64 lung POCUS video recordings, divided into 39 videos of COVID-19, 14 of (typical bacterial) pneumonia and 11 of healthy patients.	https://github.com/jannisborn/covid19_pocus_ultrasound (29 November 2020)
Enlarged POCUS dataset [60]	139 recordings (106 videos + 33 images) with convex or linear probes. 63 COVID-19, 34 bacterial pneumonia, 7 virial pneumonia and 35 healthy cases.	https://github.com/jannisborn/covid19_pocus_ultrasound/tree/master/data (29 November 2020)
New POCUS dataset [61]	202 videos and 59 images from 216 patients. COVID-19, bacterial pneumonia, non-COVID-19 viral pneumonia and healthy controls.	https://github.com/BorgwardtLab/covid19_ultrasound (1 December 2021)
ICLUS-DB [35]	30 cases of confirmed COVID-19 for a total of about 60,000 frames by the time of publishment.	https://covid19.disi.unitn.it/iclusdb (29 November 2020)
Extended ICLUS-DB [62]	An extended and fully annotated version of *ICLUS-DB*. 277 LUS videos from 35 patients (17 positive COVID-19, 4 COVID-19 suspected and 14 healthy patients).	https://iclus-web.bluetensor.ai (29 November 2020)
COVIDx-US [63]	59 COVID-19 videos, 37 non-COVID-19 videos, 41 videos with other lung diseases/conditions, and 13 videos of normal patients.	https://github.com/nrc-cnrc/COVID-US (1 December 2021)

**Table 2 jimaging-08-00065-t002:** Summary of research articles on AI applications of LUS for COVID-19.

Articles	Time	Datasets	Techniques	Main Tasks	Results
Born et al. [59]	May 2020	POCUS dataset [59]: 64 videos (39 COVID-19, 14 bacterial pneumonia, and 11 healthy controls)	VGG16	Classifying frames/videos as COVID-19, bacterial pneumonia, or healthy.	* AUC: 0.94 Accuracy: 0.89 Sensitivity: 0.96 Specificity: 0.79 F1-score: 0.92
Roy et al. [62]	August 2020	35 patients (17 COVID-19, 4 COVID-19 suspected, and 14 healthy controls)	Spatial Transformer Networks (STN) & U-Net	Scoring frames/videos; Segmenting COVID-19 imaging biomarkers.	Accuracy: 0.96 Recall: 0.6 ± 0.07 Precision: 0.7 ± 0.19 F1-score: 0.61 ± 0.12
Horry et al. [102]	August 2020	Multimodal dataset of X-ray, ultrasound, and CT (COVID-19, pneumonia, and Normal)	VGG16/19, ResNet50, Inception V3, Xception, InceptionResNetV2, NASNet, and DenseNet121	Classifying COVID-19, pneumonia, and normal cases with limited datasets.	Recall: 1.0 Precision: 1.0 F1-score: 1.0
Born et al. [60]	September 2020	139 recordings (63 COVID-19, 41 non-COVID-19 pneumonia, and 35 healthy controls)	VGG16	Classifying COVID-19 US videos; Localizing spatio-temporally pulmonary biomarkers.	AUC: 0.94 ± 0.03 Recall: 0.98 ± 0.04 Specificity: 0.91 ± 0.08 Precision: 0.91 ± 0.08 MCC: 0.89 ± 0.06 F1-score 0.94 ± 0.04
Hou et al. [86]	October 2020	2800 images (740 A-line, 1150 B-line and 910 consolidation images)	Adjusted Bias (Saab) multilayer network	Classifying consolidation vs A-line vs B-line.	Accuracy: 0.97
Roberts et al. [74]	November 2020	POCUS dataset [59]	VGG16 & ResNet18	Classifying COVID-19, bacterial pneumonia, and control cases.	Accuracy: 0.86 AUC: 0.90
Carrer et al. [66]	November 2020	Subsets of the *ICLUS-DB* database [66]: 29 cases (10 negatives, 15 positives, and four suspected COVID-19)	SVM	Detecting pleural line automatically; Scoring LUS images.	Accuracy: 0.85–0.98 Sensitivity: 0.85–0.93 Specificity: 0.95–0.99
Liu et al. [95]	November 2020	71 patients with 6836 images sampled from 678 videos	ResNet50	Classifying A-line, B-line, pleural lesion, and pleural effusion.	Accuracy: 0.98 Sensitivity: 0.99 Specificity: 0.92
Baloescu et al. [93]	November 2020	2415 subclips rated for severity of B-lines, from 0 (none) to 4 (severe)	Custom-designed CNNs	Detecting B-lines from LUS clips to evaluate COVID-19 severity.	AUC: 0.97 Sensitivity: 0.81–0.98 Specificity: 0.84–0.99 Kappa: 0.79–0.97
Che et al. [90]	February 2021	POCUS dataset and ICLUS-DB: 51 COVID-19, 13 pneumonia, and 12 healthy subjects	ResNet	Classifying COVID-19 from LUS data.	Accuracy: 0.95 Recall: 0.99 Precision: 0.96 F1-score: 0.9
Muhammad et al. [89]	February 2021	121 videos (45 for COVID-19, 23 for bacterial pneumonia, and 53 for healthy); 40 images (18 for COVID-19, 7 for bacterial pneumonia, and 15 for healthy)	ResF module	Classifying COVID-19, bacterial pneumonia, and healthy cases.	AUC: 0.99 Accuracy: 0.92 Recall: 0.93 Precision: 0.92
Dastider et al. [88]	February 2021	ICLUS-DB: 58 videos (38 with a convex probe, and 20 with a linear probe) scored based on a 4-level scoring system	DenseNet-201	Scoring LUS images.	Accuracy: 0.79 ± 0.06/0.68 ± 0.03 Sensitivity: 0.79 ± 0.06/0.68 ± 0.03 Specificity: 0.90 ± 0.03/0.77 ± 0.14 F1-score: 0.79 ± 0.06/0.67 ± 0.03
Arntfield et al. [97]	February 2021	243 patients (81 hydrostatic pulmonary edema (HPE), 78 non-COVID ARDS (NCOVID), and 84 COVID-19)	Xception	Classifying COVID-19, NCOVID and HPE pathologies.	AUC: 0.97 Sensitivity: 0.92 Specificity: 0.88 Precision: 0.71 F1-score 0.81
Tsai et al. [77]	March 2021	70 patients (39 abnormal and 31 normal)	STN	Classifying normal vs pleural effusion classes.	Accuracy: 0.92 Recall: 0.88 F1-score: 0.9
Hu et al. [100]	March 2021	Multicenter and multimodal ultrasound data from 104 patients	ResNeXt	Scoring lung sonograms based on classifications of pathology indicators.	Accuracy: 0.94 Sensitivity: 0.76 Specificity: 0.96 Precision: 0.82
Xue et al. [98]	April 2021	313 patients classified into four types (mild, moderate, severe, and critical severe)	VGG	Classifying severity of COVID-19 patients from LUS and clinical information.	Accuracy: 0.88 Recall: 0.85 Precision: 0.8 F1-score: 0.87
Gare et at. [84]	April 2021	Four patients (three COVID-19 positives and one control)	U-net	Segmenting A-line, B-line, and pleural line; Classifying normal vs. pneumonia vs. COVID-19.	Accuracy: 0.85 Recall: 0.91 Precision: 0.89 F1-score: 0.90
Mento et al. [78]	May 2021	1488 videos from 82 patients, scored 0-3 scales	STN & U-Net and DeepLab v3+	Scoring LUS videos.	Accuracy: 0.86
Yaron et al. [76]	June 2021	35 patients (17 COVID-19, 4 COVID-19 suspected, and 14 healthy controls)	Resnet18	Scoring LUS frames.	F1-score: 0.69
Raghavi et al. [71]	June 2021	765 images (266 positive COVID-19 and 499 negative cases)	ANN	Classifying a LUS dataset.	Accuracy: 0.84
Awasthi et al. [85]	June 2021	*POCUS* dataset: 64 videos (11 healthy, 14 pneumonia, and 39 COVID-19 patient)	MobileNet	Classifying COVID-19, bacterial pneumonia, and healthy cases.	Accuracy: 0.83 Sensitivity: 0.92 Specificity: 0.71 Precision: 0.83 F1-score: 0.87
Zheng et al. [99]	June 2021	Multimodal dataset: 1393 doctor–patient dialogues and 3706 images for COVID-19 patients; and 607 dialogues and 10,754 images for non-COVID-19 patients	Temporal NN	Classifying COVID-19 vs. non-COVID-19 casese.	Accuracy: 0.98 Sensitivity: 0.99 Specificity: 0.99 Precision: 0.99 AUC: 0.99 F1-score: 0.99
Sadik et al. [91]	July 2021	POCUS dataset [59]	DenseNet-201, ResNet-152V2, Xception, VGG19, and ImageNet	Classifying COVID-19, pneumonia, and normal cases.	Accuracy: 0.91 Sensitivity: 0.91 Specificity: 0.90 F1-score: 0.90
Barros et al. [87]	August 2021	185 videos (69 COVID-19, 50 bacterial pneumonia, and 66 healthy controls)	POCOVID-Net, DenseNet, ResNet, Xception, and NASNet	Classifying COVID-19, pneumonia, and normal cases.	Accuracy: 0.91–0.93 Recall: 0.84-0.97 Specificity: 0.90–1.0 Precision: 0.89–1.0 F1-score: 0.86–0.95
Diaz-Escobar et al. [73]	August 2021	3326 images (1283 for COVID-19, 731 for bacterial pneumonia, and 1312 for healthy controls)	VGG19, InceptionV3, Xception, and ResNet50	Classifying COVID-19, pneumonia, and normal cases.	AUC: 0.97 ± 0.01 Accuracy: 0.89 ± 0.02 Recall: 0.86 ± 0.03 F1-score: 0.88 ± 0.03 Precision: 0.9 ± 0.03
Ebadi et al. [81]	August 2021	300 patients (100 for each ARDS feature: A-line, B-line, and consolidation)	3D ConvNet	Classifying A-line, B-line, and consolidation and/or pleural effusion from videos.	AUC: 0.91–0.96 Accuracy: 0.9 Recall: 0.86–0.92 Precision: 0.93–0.98 F1-score: 0.87–0.94
La Salvia et al. [105]	August 2021	450 patients (278 positive and 172 negative cases)	ResNet18, ResNet50	Classifying four/seven classes of LUS.	AUC: 0.98–1.0 Accuracy: 0.98–1.0 Recall: 0.97–0.99 Precision: 0.98–0.99 F1-score: 0.97–0.99
Panicker et al. [94]	September 2021	5000 images from seven subjects (1000 images per class)	VGG16	Detecting pleura and generating acoustic features; Classifying five classes of LUS images.	Accuracy: 0.97 Sensitivity: 0.92 Specificity: 0.98
Mento et al. [79]	September 2021	100 patients with 133 LUS exams scored to four levels	STN & U-Net and DeepLab v3+	Scoring LUS videos.	Accuracy: 0.82
Al-Jumaili et al. [70]	October 2021	2995 images (988 COVID-19, 731 pneumonia, and 1276 regular images, available on Kaggle)	SVM & Resnet18, Resnet50, GoogleNet, and NASNet-Mobile	Detecting pathology features from LUS images; Classifying COVID-19, pneumonia, and regular cases.	Accuracy: 0.99 Sensitivity: 0.99 Specificity: 0.99 F1-score: 0.99
Karnes et al. [104]	October 2021	13103 normal, 4900 pneumonia, and 8633 COVID-19 frames	LDA & MobileNet	Classifying COVID-19, pneumonia, and healthy cases.	AUC: 0.95
Demi et al. [80]	December 2021	220 patients (100 positive patients and 120 post-COVID-19 patients)	STN & U-Net	Testing protocols for grading LUS.	Accuracy: 0.80
Roshankhah et al. [82]	Decemberc 2021	32 patients (14 confirmed COVID-19, 4 suspected cases and 14 controls)	U-Net	Scoring severity in 4-scale stages; Investigating the impact of various training/test splitting schemes.	Accuracy: 0.95/0.75
Wang et al. [69]	January 2022	27 cases (13 moderate, seven severe, and seven critical cases of COVID-19)	SVM	Scoring the severity of COVID-19 pneumonia by pleural line and B-lines.	AUC: 0.88–1.0 Sensitivity: 0.93 Specificity: 1.0
Durrani et al. [83]	July 2022	28 patients (10 unhealthy and 18 healthy)	STN & U-Net	Detecting Consolidation/Collapse in LUS videos/frames.	AUC: 0.73 ± 0.3 Accuracy: 0.89 ± 0.16 Recall: 0.84 ± 0.23 Precision: 0.59 ± 0.28 F1-score: 0.67 ± 0.25

* Area under curve (AUC).

## Data Availability

Not applicable.

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
