# Peer review of "Review of Machine Learning in Lung Ultrasound in COVID-19 Pandemic"

_2313-433X, 2022, doi:10.3390/jimaging8030065_

Round 1

Reviewer 1 Report

Paper: Review of Machine Learning in Lung Ultrasound in COVID-19 Pandemic

Author: Wang et al.

Venue: Journal of Imaging - MDPI

Review Date: Feb 23, 2022

This paper provides a systematic review of AI methods for LUS / POCUS related to covid-19. Within this review is included LUS scan protocols, features in LUS images that are indicative of covid-19, as well as recent developments in databases and AI methods related to covid-19. 

This paper has a clear structure and is well written and very comprehensive. The authors have done a good job of setting the context of POCUS and discussing implications of deep learning methods within the clinical setting. The citations are recent and credible.

I suggest this paper is accepted with minor revisions:

Revision 1

I think adding a “task” column to Table 2 would greatly strengthen the paper. Currently the ACC, SENS, SPEC, AUC, and F1-score are being reported for a variety of papers all working on LUS covid applications, making it seem like a one-to-one comparison can be made between these papers. However, the task of these papers range vastly, making it not able to directly compare these metrics. For example:

  • [98] is building classifiers to compare “(1) COVID-19 (COVID), non-COVID acute respiratory distress syndrome (NCOVID) or (3) hydrostatic pulmonary edema (HPE).”
  • [87] is classifying “consolidation vs a-line vs b-line” (not mentioned in paper -- had to look at original paper).
  • [84] is conducting “Consolidation/Collapse Classification”
  • [59] is classifying “(COVID-19 pneumonia, bacterial pneumonia, and healthy controls)”

Adding these high level details to Table 2 would add clarity and provide a helpful summary when reading the document.

Revision 2

The “AI Architecture” column in Table 2 should be more consistent. Sometimes broadly “CCNs” are stated, while other times specific architecture are mentioned (e.g. “ResNet18”, “U-Net”, “VGG”, etc.). All these specific architectures are types of “CNNs”. Providing more clarity on this will be complementary to adding the “task” being done (see Revision 1). I suggest mentioning the specific architectures opposed to broadly saying “CNNs”.

Revision 3

There are minor spelling and grammar errors in the paper. Some that I found were:

  • Line 135: “CORVID-19”
  • Line 279: extra comma after “Net ,”
  • In Table 2:
    • Carrer et al. [66] has extra ‘-’ after the reported values.
    • Ebadi et al. [82] has extra ‘-’ after the reported values.

End of Review

Reviewer 2 Report

I enjoyed reading this review manuscript, which provides the informative and updated knowledge regarding machine learning approaches for pulmonary ultrasound images of COVID-19.

This review manuscript should be accepted for broad range of audiences including medical practitioners and scientists in medical image analysis. This is well documented review paper.

Only minor point is that in Table 2, it would be better to add information regarding the "Datasets" more detailly if possible (because audiences can not clearly understand the distribution of dataset cases in the current manuscript).

In addition, in Table 2, if applicable, it would be best to add confidence intervals (if authors can pick up from the literatures). 
